# Impact of the Gram-Negative-Selective Inhibitor MAC13243 on In Vitro Simulated Gut Microbiota

**DOI:** 10.3390/ph15060731

**Published:** 2022-06-09

**Authors:** Frida Svanberg Frisinger, Mattia Pirolo, Duncan Y. K. Ng, Xiaotian Mao, Dennis Sandris Nielsen, Luca Guardabassi

**Affiliations:** 1Department of Veterinary and Animal Sciences, Faculty of Health and Medical Sciences, University of Copenhagen, 1870 Frederiksberg, Denmark; frida.frisinger@sund.ku.dk (F.S.F.); mapi@sund.ku.dk (M.P.); yong.kai.ng@sund.ku.dk (D.Y.K.N.); 2Department of Bacteria, Parasites and Fungi, Statens Serum Institut, 2300 Copenhagen, Denmark; 3Department of Food Science, Faculty of Science, University of Copenhagen, 1870 Frederiksberg, Denmark; xiaotian.mao@food.ku.dk (X.M.); dn@food.ku.dk (D.S.N.)

**Keywords:** *Escherichia coli*, antibiotic targets, selective antimicrobial activity, LolA, microbiome

## Abstract

New Gram-negative-selective antimicrobials are desirable to avoid perturbations in the gut microbiota leading to antibiotic-induced dysbiosis. We investigated the impact of a prototype drug (MAC13243) interfering with the Gram-negative outer membrane protein LolA on the faecal microbiota. Faecal suspensions from two healthy human donors were exposed to MAC13243 (16, 32, or 64 mg/L) using an in vitro gut model (CoMiniGut). Samples collected 0, 4, and 8 h after exposure were subjected to viable cell counts, 16S rRNA gene quantification and V3-V4 sequencing using the Illumina MiSeq platform. MAC13243 exhibited concentration-dependent killing of coliforms in both donors after 8 h. Concentrations of ≤32 mg/L reduced the growth of aerobic bacteria without influencing the microbiota composition and diversity. An expansion of Firmicutes at the expense of Bacteroidetes and Actinobacteria was observed in the faecal microbiota from one donor following exposure to 64 mg/L of MAC13242. At all concentrations tested, MAC13243 did not lead to the proliferation of *Escherichia coli* nor a reduced abundance of beneficial bacteria, which are typical changes observed in antibiotic-induced dysbiosis. These results support our hypothesis that a drug interfering with a specific target in Gram-negative bacteria has a low impact on the commensal gut microbiota and, therefore, a low risk of inducing dysbiosis.

## 1. Introduction

Antibiotics are essential medicines for reducing the burden of bacterial diseases. However, treatment with broad-spectrum antibiotics can damage the commensal gut microbiota, resulting in complications that range from colonization with multidrug-resistant bacteria to life-threatening infections caused by *Clostridioides difficile* [1]. This phenomenon, usually referred to as antibiotic-induced dysbiosis, is caused by the proliferation of opportunistic pathogens accompanied by the killing of beneficial bacteria that would otherwise provide protection for the host [2]. The human gut microbiota imparts specific functions in the host, including the development and maintenance of the immune system, and the protection from pathogen expansion [3]. Although the gut microbiota exhibits a remarkable resilience towards changes both over short and long time periods, antibiotics have a tremendous impact on the bacterial community and can lead to persisting effects on the gut microbiota [4,5].

One approach to circumvent antibiotic-induced dysbiosis is the development of narrower-spectrum antibacterial drugs that are selective for a target pathogen and have a limited impact on the commensal microbiota [6]. We previously performed an in silico study to identify novel antimicrobial drug targets selective for *Escherichia coli* and *Klebsiella pneumoniae* that are absent in important beneficial bacteria residing in the gut [7]. One of the identified targets, LolA, is a key component of the *E. coli* outer membrane biogenesis machinery [8]. This protein is the target of a commercially available compound (MAC13243) that belongs to a new chemical class and is a promising antibacterial lead candidate with Gram-negative-selective properties [9]. However, the effects of this compound on the gut microbiota are not yet established. Here, we assess the impact of MAC13243 on the gut microbiota composition and diversity using the CoMiniGut, an in vitro system simulating the colon passage in the human gut [10]. The purpose of this study is to provide an initial proof-of-concept supporting our hypothesis that a Gram-negative-selective drug interfering with a specific target in *E. coli* has a limited impact on the commensal gut microbiota.

## 2. Results

### 2.1. Drug Impact on Coliform and Total Viable Cells

To enumerate the number of viable aerobically culturable cells, bacterial counts on coliform-selective agar (MacConkey) and non-selective agar (Brain Heart Infusion) were performed on faecal samples from two infant donors exposed to three concentrations of MAC1243 (16, 32, and 64 mg/L) using untreated samples as negative controls. MAC13243 exhibited a dose-dependent killing of coliforms, and counts fell below the detection limit after 8 h of exposure (Figure 1A,C). Regrowth of the coliform population from donor 2 was observed after 8 h regardless of the drug concentration (data not shown). To evaluate whether the regrowth was associated with higher levels of resistance in the coliform population of donor 2, the MICs of MAC1243 were determined in five *E. coli* isolates collected from each donor after 24 h of incubation. Indeed, four out of five isolates from donor 2 showed a higher MIC (256 mg/L) compared to those from donor 1 (4–8 mg/L). A slight reduction in the growth of the total aerobically culturable bacteria was observed in the treated samples from both donors as compared to the untreated control (Figure 1B,D). However, quantitative PCR (qPCR) quantification of the 16S rRNA gene showed no changes in the total number of bacteria between the treated and control samples from both donors (Figure 2). This discrepancy could be due to the detection of the 16S rRNA gene from dead cells in the treated samples by qPCR.

### 2.2. Drug Impact on Microbiota Composition and Diversity

The bacterial compositions of the faecal samples were assessed using 16S rRNA gene sequencing. The microbiota compositions differed between the two faecal samples, which presented significantly different bacterial compositions based on the Bray–Curtis dissimilarity matrix (PERMANOVA, *p* = 0.001) (Figure 3). Such a difference was expected, as faecal microbiota are known to display high interindividual variation, which is largely attributed to differences in the host’s lifestyle and environmental factors [11].

The exposure of the faecal microbiota from donor 1 to concentrations below 64 mg/L did not result in any substantial changes in the relative abundance of taxa compared to the untreated control, and all samples exhibited a reduction in Firmicutes and Bacteroidetes, accompanied by a temporal increase in Actinobacteria (Figure 4A). In contrast, treatment with 64 mg/L of MAC13243 led to a gradual increase in the relative abundance of Firmicutes at the expense of Bacteroidetes and Actinobacteria. In the faecal microbiota of donor 2, the most apparent change was an increase in Proteobacteria in the untreated control. This change became especially obvious after 8 h (Figure 4A) but was prevented by all of the MAC13243 concentrations tested. These results are in agreement with the bactericidal effect of MAC13243 on coliforms in the treated samples and with the relative increase in viable coliforms observed in the untreated faecal microbiota of this donor (Figure 1C).

Microbial richness, expressed by the Chao1 index, decreased over time in the control samples of both donors (Figure 4B). A relative increase in the number of bacterial species compared to the untreated control occurred after 4 h of exposure to 16–32 mg/L of MAC13243 for donor 1, and at all drug concentrations and exposure times for donor 2. A relative decrease was only detected for donor 1 after 4 h of exposure to 64 mg/L and after 8 h of exposure to any of the three drug concentrations tested (Figure 4B).

At the genus level, the increase in Firmicutes seen in the sample from donor 1 following treatment with 64 mg/L of MAC13243 was related to an increase in *Clostridium sensu stricto* 1 (Figure 4C). *Clostridium sensu stricto* 1 contains non-pathogenic members of the gut microbiota, such as *Clostridium butyricum*, a proposed probiotic species capable of antagonising pathogens, including *E. coli* [12,13]. *C. difficile*, the major pathogen within the *Clostridium* cluster XI [14], was not identified in either donor. The untreated faecal microbiota of donor 2 became dominated by *Escherichia* after 8 h (Figure 4C), which is in agreement with the increase in Proteobacteria observed at the phylum level (Figure 4A) as well as with the increase in coliforms detected by viable cell counts on MacConkey agar (Figure 1C).

To investigate the degree of conservation of the MAC13243 target LolA among Gram-negative bacteria, phylogenetic distances between the LolA protein sequences of members from both Proteobacteria and Bacteroidetes were inferred. The analysis showed that Proteobacteria formed a discrete clade clearly distinct from all members of Bacteroidetes (Figure 5). LolA is highly conserved among *Enterobacteriaceae*, which formed a separate cluster (supported by a >70% bootstrap value) from all other members of Proteobacteria (Figure 5).

## 3. Discussion

Antibiotic-induced dysbiosis is normally associated with reduced microbiome diversity and richness, reduced abundance of beneficial bacteria, such as *Bifidobacterium*, and an increased abundance of Proteobacteria, including the opportunistic pathogen *E. coli* [15]. Our study shows a selective antimicrobial activity of MAC13243 on Proteobacteria and *E. coli* compared to the relatively low impact on Bacteroidetes and Firmicutes, which represent the main phyla of Gram-negative and Gram-positive commensal bacteria in the gut, respectively. Furthermore, the drug did not affect the relative abundance of *Bifidobacterium* and even increased the presence of these beneficial bacteria in the faecal microbiota of one of the two donors. Collectively, these data strongly suggest that exposure to MAC13243 did not induce dysbiosis due to the selective inhibition of *E. coli*, largely leaving the surrounding microbiota intact.

The effects of MAC13243 on microbial richness differed between the two donors and were influenced by the drug concentration and exposure time. Overall, the richness either remained stable (donor 1) or increased (donor 2) following exposure to MAC13243 concentrations of 16 mg/L for 8 h, indicating that exposure of the gut microbiota to this drug concentration does not result in a reduction in the microbial diversity, which is another key parameter to evaluate the risk of dysbiosis. The marked reduction in the microbial richness observed in donor 1 after 8 h of exposure to 64 mg/L (Figure 4B) was associated with a proliferation of Firmicutes at the expense of Actinobacteria and Bacteroidetes (Figure 4A), which in turn was largely associated with a proliferation of Clostridia (Figure 4C). The different impacts of high concentrations of MAC13243 on microbial diversity observed between the two donors may reflect differences in the initial microbiota compositions of their faeces. It is well-known that the gut microbial composition varies among individuals and is influenced by various factors, including the host’s genetics and lifestyle [11]. As the two faecal samples analysed in this study did not contain *C. difficile*, further research is needed to assess whether high concentrations of MAC13243 may lead to the selection of this opportunistic enteric pathogen.

The difference in MICs observed amongst *E. coli* isolates from the two donors indicates that a resistant phenotype already exists in nature, even though MAC13243 has never been used in clinical practice. The high MIC (256 mg/L) observed in some of the isolates is not a favourable trait for the clinical use of MAC13243 as a systemic antimicrobial drug. Notably, this is just a proof-of-concept study to validate the hypothesis that a drug interfering with specific targets in *E. coli* has a limited impact on the commensal gut microbiota. For different reasons, the results of the study are far from demonstrating that MAC13243 is a valuable drug lead for antimicrobial chemotherapy. More research is needed to determine the MIC distribution of this compound in clinical *E. coli* isolates and to assess whether such high drug concentrations can be achieved at the infection site without causing any side effects to the host.

We acknowledge some limitations of our study. The main limitation is the limited sample size, which prevents any definitive conclusions on the clinical potential of MAC13243. The use of only two donors prevents any generalized conclusions about the impact of MAC13243 on the gut microbiota. Additionally, no comparison with a clinically relevant broad-spectrum antibiotic was performed in the present study, which would provide insight into the relative specificity of MAC13243. However, the study provides an initial proof-of-concept supporting the hypothesis that a drug interfering with specific targets in *E. coli* has a limited impact on the commensal gut microbiota. Obviously, further validation is required to translate this concept from the laboratory to clinical practice, and the results of the study suggest that MAC13243 may not be the best drug candidate to further explore this innovative antimicrobial treatment strategy. The young age of the two donors is another limitation of the study since the infant gut microbiota is variable in its composition and less stable over time [16]. Accordingly, a follow-up study utilizing a larger number of volunteers and including adult individuals from different backgrounds is required to confirm our initial observations and to provide further insights into the selective antimicrobial activity of MAC13243. Moreover, such a study should include an antibiotic with a known impact on gut microbiota as a positive control.

## 4. Materials and Methods

### 4.1. Experimental Setup and Bacterial Counts

The effects of MAC13243 on the gut microbiota were evaluated using the CoMiniGut. The system was inoculated with faecal samples collected from two healthy male infant volunteers not exposed to antibiotics six months prior to sampling, as previously described (Ethical Committee of the Capital Region of Denmark registration number H-20028549) [10]. Briefly, faecal samples were diluted 1:5 in PBS, vortexed, and 3 mL of the faecal slurry was mixed with 27 mL of basal colon medium (BCM) [10]. Stirred anaerobic fermentation vessels were set up and aseptically filled with 5-mL aliquots of the BCM supplemented with faecal slurry for continuous culture. All procedures prior CoMiniGut incubation were performed in an anaerobic chamber. For each donor, four fermentations were run in parallel and included three concentrations of MAC13243 (16, 32, and 64 mg/L), based on the previously reported MIC of 16 mg/L in *E. coli* [9], and a negative control. One mL was collected from each faecal suspension immediately prior to drug inoculation and after 4 and 8 h of incubation at 37 °C. Of this 1 mL, 900 µL were immediately frozen at −80 °C. The remaining 100 µL was serially diluted (1:10) and 5 µL were spotted in triplicates on Brain Heart Infusion agar (Oxoid, Roskilde, Denmark) and MacConkey agar (Becton Dickinson, Albertslund, Denmark) for the total counts of culturable aerobic bacteria and coliforms, respectively, after 24 h of incubation at 37 °C under aerobic conditions. Both the total culturable bacteria and coliforms were further counted after 24 h of exposure (experimental endpoint). The MICs of MAC13243 were determined according to the CLSI recommendations [17] on five *E. coli* isolates from each untreated faecal sample after species confirmation by MALDI-TOF (Vitek MS, bioMerieux, Marcy-l'Étoile, France) (see Appendix A for further details).

### 4.2. 16S rRNA Gene Quantification and Sequencing

DNA was extracted from the defrosted samples using the QIAamp Fast DNA Stool Mini kit (QIAGEN, Hilden, Germany) with a bead-beating step, as previously described [18]. The total bacteria were quantified by qPCR targeting the 16S rRNA gene [19]. The microbiota compositions were estimated by sequencing the V3-V4 region of the 16S rRNA. Libraries were prepared using the Quick-16S NGS Library Prep Kit (Zymo Research, Irvine, CA, USA) and sequenced using the MiSeq Reagent Kit v3 (600 cycles, 2 × 300 bp paired-end reads). Sequencing data were processed using DADA2 v1.14.1 as implemented in R v3.6.1. The 16S rRNA sequencing data have been submitted to the NCBI Sequence Read Archive (SRA) under BioProject PRJNA772613.

## 5. Conclusions

MAC13243 inhibited *E. coli* growth without significantly reducing the microbial diversity and relative abundance of beneficial bacteria under in vitro conditions mimicking the colon passage in the human gut. Such properties are favourable for preventing antibiotic-associated dysbiosis, suggesting that it is possible to develop drugs targeting *E. coli* with minimal impact on the commensal gut microbiota. However, these properties should be confirmed using a larger number of individuals and an antibiotic with a known impact on gut microbiota as a positive control.

## Figures and Tables

**Figure 1 pharmaceuticals-15-00731-f001:**
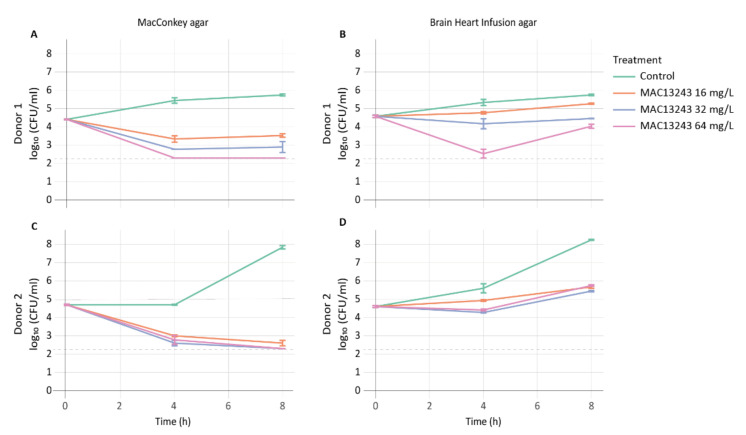
Viable cell counts of coliforms on MacConkey agar (**A**,**C**), and of total bacteria on Brain Heart Infusion agar (**B**,**D**) in faecal samples from two human donors after exposure to MAC13243. The limit of detection (200 CFU/mL) is highlighted with a dashed line.

**Figure 2 pharmaceuticals-15-00731-f002:**
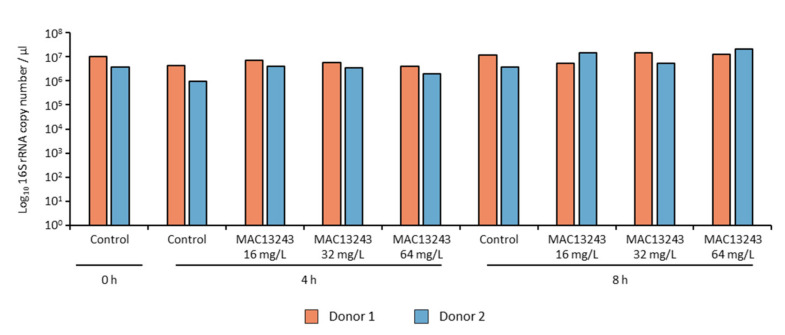
Quantification of total bacteria populations using qPCR of the 16S rRNA gene.

**Figure 3 pharmaceuticals-15-00731-f003:**
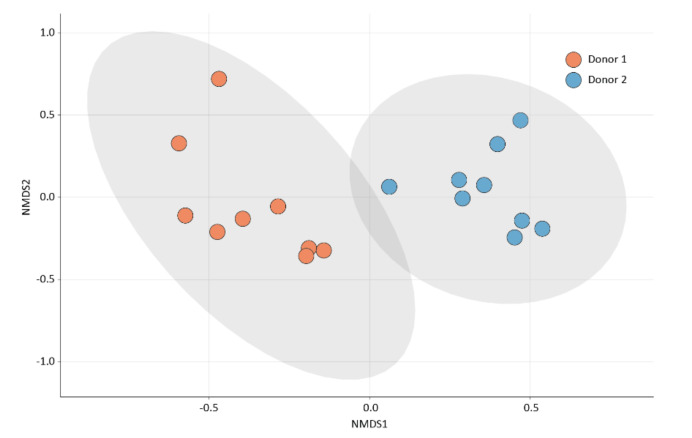
Two-dimensional non-metric multidimensional scaling (nMDS) plot of the microbial community compositions in the donor samples. An nMDS plot based on the Bray–Curtis dissimilarity matrix was used to simultaneously visualise individual samples (dots) originating from donor 1 (D1, red) and donor 2 (D2, blue). Sample clustering of the two donors was significantly different (PERMANOVA, *p* = 0.001).

**Figure 4 pharmaceuticals-15-00731-f004:**
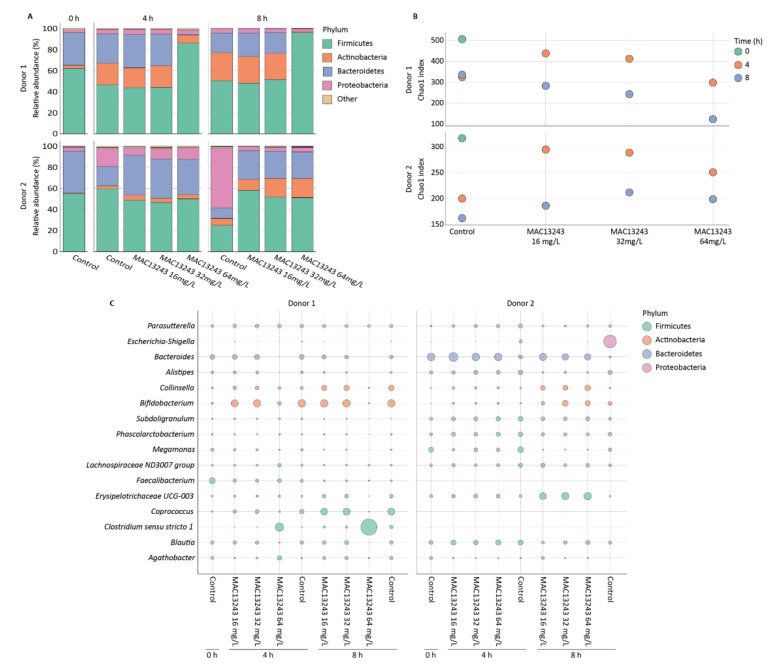
Effects of MAC13243 on microbiota composition and diversity in faecal samples from two human donors; (**A**) relative abundance at the phylum level; (**B**) alpha diversity; (**C**) bubble plots showing the proportion of the highest abundant genera across all samples.

**Figure 5 pharmaceuticals-15-00731-f005:**
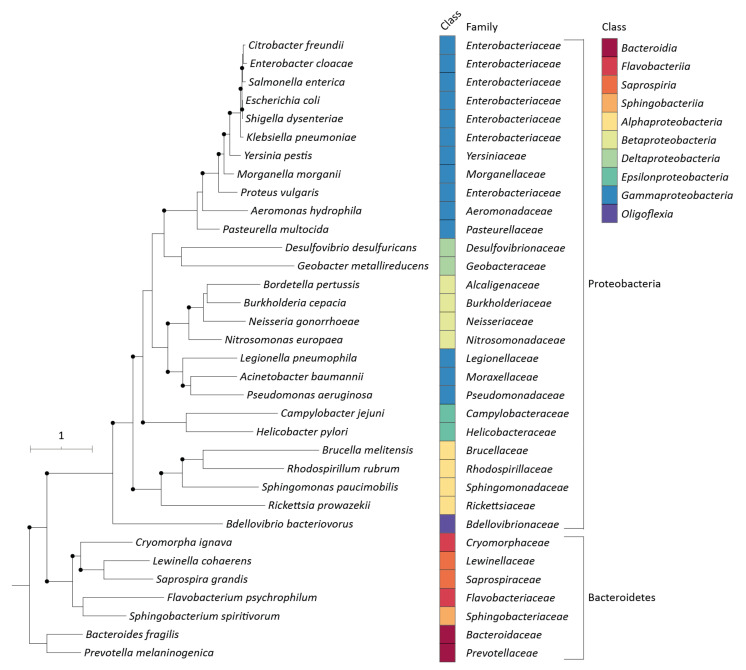
Rooted tree based on the alignment of the LolA protein sequences of the main Gram-negative bacteria. Bootstrap values of >70% (1000 replicates) are illustrated by black, filled circles at the nodes. The scale bar indicates the expected number of substitutions per site.

## Data Availability

Data is contained within the article and Appendix A. Sequencing data can be accessed from the NCBI Sequence Read Archive (SRA) under BioProject PRJNA772613.

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
