# Peer review of "Impact of the Gram-Negative-Selective Inhibitor MAC13243 on In Vitro Simulated Gut Microbiota"

_pharmaceuticals, 2022, doi:10.3390/ph15060731_

Round 1

Reviewer 1 Report

In the article "Impact of the Gram-negative-selective inhibitor MAC13243 on in vitro simulated gut microbiota", the authors describe the use of the Gram-negative specific drug MAC13243 to treat the fecal slurries from two healthy donors in order to observe the effects of the drug on the commensal population. They find that the drug appears to have targeted effects in the sample from one donor, but not the other.

I cannot support publishing this paper in its current form. As mentioned by the authors in the discussion, there are a number of issues. The first is the number of donors. Given that this drug was only potentially promising in one sample and not the other, it really offers no interest to readers. Additionally, there are no new methods presented here that would justify publishing negative data. My suggestions are to try spiking healthy samples with a known number of coliphage to measure, specifically, the decrease caused by the drug. Alternatively, an increased number of donors (healthy and sick) would substantially improve the value of the assessment of targeted antibiotics on the commensal populations.

Author Response

Comment 1.1. In the article "Impact of the Gram-negative-selective inhibitor MAC13243 on in vitro simulated gut microbiota", the authors describe the use of the Gram-negative specific drug MAC13243 to treat the fecal slurries from two healthy donors in order to observe the effects of the drug on the commensal population. They find that the drug appears to have targeted effects in the sample from one donor, but not the other.

I cannot support publishing this paper in its current form. As mentioned by the authors in the discussion, there are a number of issues. The first is the number of donors. Given that this drug was only potentially promising in one sample and not the other, it really offers no interest to readers. Additionally, there are no new methods presented here that would justify publishing negative data. My suggestions are to try spiking healthy samples with a known number of coliphage to measure, specifically, the decrease caused by the drug. Alternatively, an increased number of donors (healthy and sick) would substantially improve the value of the assessment of targeted antibiotics on the commensal populations.

Reply 1.1. The authors thank the reviewer for the constructive criticism, which has contributed to improve the quality of the manuscript. The reviewer’s statement that “this drug was only potentially promising in one sample and not the other” is probably due to lack of clarity in the way the results were presented and discussed in the original manuscript. Accordingly, we have revised the manuscript to avoid misunderstanding on this point. Please note that the number of viable coliforms decreased in both faecal samples exposed to any concentrations of MAC13243, whereas an increase was observed in the untreated control samples (Figure 1). The selective activity of this drug against E. coli was confirmed by 16S rRNA sequencing, which also showed survival of beneficial bacteria such as Bifidobacterium and absence of any significant reduction of microbial diversity at drug concentrations of 16-32 mg/L in both samples (Figure 3). Thus, although we agree with the reviewer that an increased number of donors would substantially improve the value of the study, the data presented in the paper are still valuable because they support our hypothesis that a drug interfering with specific targets in E. coli has limited impact on the commensal gut microbiota. Unfortunately, performance of additional work is not possible because the the EU project (CARTNET) supporting this work ended last year.

Reviewer 2 Report

The manuscript entitled “Impact of the Gram-negative-selective inhibitor MAC13243 on in vitro simulated gut microbiota” is well written and well-structured especially in the conceptualization and implementation of the experimental phase.

The limitations of the study are well highlighted in the conclusions by the Authors themselves, who emphasize the need for a greater number of samples and especially the availability of an antibiotic as a positive control as an effect on the microbiota.

Thus, I find no flaws in the study except for a greater description of these decidedly limiting aspects for the generalization of the results to which I would devote more space in the discussion rather than limiting them to a brief mention in the conclusion section.

Another important consideration: if the journal would allow it, I would insert the very interesting data of the supplementary material in the body of the manuscript because they are useful to the full understanding of the work done by the Authors and more usable by the reader.

Considering the importance and topicality of the topic investigated by the Authors, I congratulate them for the scientific rigor with which they conducted the experimentation and I hope for a rapid publication of the manuscript.

Author Response

Comment 2.1. The manuscript entitled “Impact of the Gram-negative-selective inhibitor MAC13243 on in vitro simulated gut microbiota” is well written and well-structured especially in the conceptualization and implementation of the experimental phase.

The limitations of the study are well highlighted in the conclusions by the Authors themselves, who emphasize the need for a greater number of samples and especially the availability of an antibiotic as a positive control as an effect on the microbiota.

Thus, I find no flaws in the study except for a greater description of these decidedly limiting aspects for the generalization of the results to which I would devote more space in the discussion rather than limiting them to a brief mention in the conclusion section.

Reply 2.1. The authors thank the reviewer for the positive comments. The Discussion has now been expanded to include these limitations (lines 191-194).

Comment 2.2. Another important consideration: if the journal would allow it, I would insert the very interesting data of the supplementary material in the body of the manuscript because they are useful to the full understanding of the work done by the Authors and more usable by the reader. Considering the importance and topicality of the topic investigated by the Authors, I congratulate them for the scientific rigor with which they conducted the experimentation and I hope for a rapid publication of the manuscript.

Reply 2.2. Following the reviewer’s suggestion, figures S1 and S2 have now been moved to the main body as figures 3 (lines 100-104) and 5 (lines 145-148), respectively.

Reviewer 3 Report

Svanberg Frisinger and collegues test the antimicrobial effects of an experimental drug directed against an OMP only present in Gram- in an artificial gut system. The article presents an interesting selective strategy to target selected pathogens without disrupting the microbiome. The discussion is well addressed, as are well described the limitations of the study.

Abstract: “Samples collected at 0, 4 and 8 h after exposure were subjected to viable cell counts, 19 16S rRNA gene quantification and sequencing” please specify which sequencing you performed.

Line 91 “whereas the coliform population from donor 2 was able to grow after 8 h regardless of the antimicrobial concentration (Figure 3B) “. This sentence might refer to Figure 3A as well.

Is there a comment referring to Fig. 3B?

Fig 1 legend: correct CFU/mla

Please shortly mention the young age of the donors, if you didn’t, at the beginning of the results’ section, as you mention it in the discussion.

Line 156 “Moreover, exposure to MAC13243 inhibited the expansion of E. coli that was observed in the untreated controls for donor 2 (Figure 4C). Collectively these data strongly suggest that exposure to MAC13243 did not induce dysbiosis due to the selective inhibition of E. coli, largely leaving the surrounding microbiota intact” please add a comment about 64mg/L, which actually correlates with a loss of Actinobacteria and Bacteroidetes in donor 1.

Author Response

Comment 3.1. Svanberg Frisinger and collegues test the antimicrobial effects of an experimental drug directed against an OMP only present in Gram- in an artificial gut system. The article presents an interesting selective strategy to target selected pathogens without disrupting the microbiome. The discussion is well addressed, as are well described the limitations of the study.

Reply 3.1: The authors thank the reviewer for the positive comments.

Comment 3.2. Abstract: “Samples collected at 0, 4 and 8 h after exposure were subjected to viable cell counts, 19 16S rRNA gene quantification and sequencing” please specify which sequencing you performed.

Reply 3.2: Done (lines 20-21).

Comment 3.3. Line 91 “whereas the coliform population from donor 2 was able to grow after 8 h regardless of the antimicrobial concentration (Figure 3B) “. This sentence might refer to Figure 3A as well.

Reply 3.3: This comment is no longer relevant because Figure 3 was removed from the manuscript. This was done to limit the total number of figures after another reviewer suggested to include two additional figures (Fig. 3 and 5 in the revised manuscript).

Comment 3.4. Is there a comment referring to Fig. 3B?

Reply 3.4: See above.

Comment 3.5. Fig 1 legend: correct CFU/mla

Reply 3.5: Done.

Comment 3.6. Please shortly mention the young age of the donors, if you didn’t, at the beginning of the results’ section, as you mention it in the discussion.

Reply 3.6: Done (line 64).

Comment 3.7. Line 156 “Moreover, exposure to MAC13243 inhibited the expansion of E. coli that was observed in the untreated controls for donor 2 (Figure 4C). Collectively these data strongly suggest that exposure to MAC13243 did not induce dysbiosis due to the selective inhibition of E. coli, largely leaving the surrounding microbiota intact” please add a comment about 64mg/L, which actually correlates with a loss of Actinobacteria and Bacteroidetes in donor 1.

Reply 3.7: Done (lines 167-170).

Round 2

Reviewer 1 Report

Dear Authors,

Thank you for your edits and rewording to focus on the benefits of using this targeted treatment as compared to the standard antibiotic treatment seen in clinic. Given that there is no availability of funds, and that you have made clear mention of where follow on research could move, I feel able to recommend your research for publication.